# OpenReview forum: "Towards Strategic Persuasion with Language Models"
_ICLR.cc/2026/Conference — ICLR 2026 Poster_

### Official Review · Reviewer_rDdu · 2025-10-31

**Soundness:** 3
**Presentation:** 2
**Contribution:** 3
**Rating:** 6
**Confidence:** 4

**Summary:**

The paper is about measuring the persuasive capabilities of LLMs. To begin with, motivated by the established theory of Bayesian Persuasion, a framework is further adopted for the case of LLMs. Focusing on opinion change tasks, a benchmark is provided for both static and dynamic settings. A reinforcement learning type of training method is also considered to further improve the capabilities. With the proposed metric, frontier LLMs are shown to have strong capabilities in strategic persuasion. The RL framework is also shown to be promising in training specific small LLM.

**Strengths:**

> **Originality**
- The paper, inspired by the Bayesian Persuasion theory, provides a scalable and systematic approach to measuring persuasion capability of LLMs.

> **Quality**
- This work incorporates a group of frontier LLMs with various sizes in the experiments for the evaluation of persuasion gains.
- There is follow-up numerical analysis on prior, semantic diversity and persuasion strategies.

**Weaknesses:**

> **Originality**
- It would be helpful if the authors clarify whether the framework in Section 2.2 are based on the terms in former literature or proposed in this work.

> **Quality**
- Line 234: Is the size of human participants (45) large enough to draw the conclusions?

> **Clarity**
- Line 148: should the expectation be taken over $\omega\sim\mu_s$?
- The distribution of posteriors, $\tau$, could be explicitly defined with more details.
- The review of Dynamic Bayesian Persuasion could benefit from providing more details, with mathematical description of the target.

> **Significance**
- According to Line 268, could the authors clarify whether the RL framework is only concerning static persuasion? If so, the contribution covered in the abstract might need a revision.

**Questions:**

- Line 191: why, compared to the introduction in Line 162-170, notations are capitalized? e.g. the state $\omega_t$ changed to $\Omega_t$
- Line 215: How is the function $g$ defined in general (if not an index retrieval)?
- What is the reason in fixing Llama-3.1-8B-Instruct (or models with similar sizes) as Receiver models? Would using other models further restrict the persuasion gains or the improvement from RL framework?

---

> ### Author Response · Authors · 2025-11-21
> **Rebuttal - Part I**
>
> We thank the reviewer for the thoughtful review! We appreciate the recognition of our framework’s originality and the value of our empirical analyses. These insights have helped us significantly strengthen the clarity and rigor of the paper. We address all points below.
>
> ### Originality of the Proposed Framework (W1)
>
> **All the components introduced in the framework in Section 2, except the background from Bayesian persuasion, are proposed in this work.** The theoretical concepts and results follows classical theory in information design, but the measurements and benchmarks are original to our work and have not appeared in any prior literature. These elements collectively address the key challenge of systematically studying the persuasive capabilities of LLMs by contributing a principled and scalable framework.
>
> ### Statistical Analysis of Human Studies (W2)
>
> **Our study has sufficient statistical power to determine whether LLM belief updates are judged reasonable by humans.** We would like to clarify that our study generated 149 independent transcript-level annotations from 45 annotators, each containing a binary judgment of belief-update direction and a 7-point proportion rating.
>
> + For belief-direction accuracy, the observed values were $\hat{p}_1 = 0.7718$, $\hat{p}_2 = 0.8255$, and $\hat{p}_3 = 0.8523$. Using binomial standard errors, the corresponding 95% confidence intervals were $[0.71, 0.84]$, $[0.76, 0.89]$, and $[0.79, 0.91]$. One-sample proportion tests against the null $p = 0.5$ yielded $z > 6.8$ and $p < 0.001$ in all rounds, demonstrating that annotators judged belief-update directions as significantly better than chance.
>
> + For belief-proportion ratings, the mean values were $\bar{x}_1 = 4.82$, $\bar{x}_2 = 4.95$, and $\bar{x}_3 = 5.05$, with 95% confidence intervals $[4.55, 5.09]$, $[4.70, 5.20]$, and $[4.80, 5.31]$. One-sample t-tests comparing each round’s mean to the benchmark value $\mu = 4$ produced $t > 6.7$ and $p < 0.001$ for all rounds, confirming that annotators consistently judged the magnitude of belief updates as reasonable.
>
> ### Clarification of Bayesian Persuasion (W3)
>
> #### Bayesian Persuasion
>
> **All expected utilities should be taken with respect to the posterior belief $\mu_s$, i.e., $\mathbb{E}_{\omega\sim\mu_s}[\cdot]$.** This aligns with the canonical Bayesian persuasion formulation in Kamenica & Gentzkow (2011), where Receiver’s optimal action is defined via expectations over $\omega \sim \mu_s$. **We have updated the notation in Section 2.1 accordingly.**
>
> **We define $\tau \in \Delta(\Delta(\Omega))$ as the probability distribution over posteriors**, following Kamenica and Gentzkow (2011)’s construction. Formally, for any posterior $\mu$, the probability assigned by $\tau$ is given by $\tau(\mu) = \sum_{\mu_s = \mu} \sum_{\omega' \in \Omega} \pi(s \mid \omega') \mu_0(\omega')$, which is the total probability under the prior and the signal structure of all realizations $s$ that lead the receiver to hold posterior $\mu$. Thus, $\tau$ provides a representation of the informational consequences of the signal by specifying both the set of posterior beliefs the signal may generate and the likelihood with which each such belief arises. **We have incorporated related formal definition in Section 2.1 accordingly.**

---

> ### Author Response · Authors · 2025-11-21
> **Rebuttal - Part II**
>
> ### Clarification of Bayesian Persuasion (W3)
>
> #### Dynamic Bayesian Persuasion
>
> **We have provided formal descriptions of dynamic Bayesian persuasion introduced by Ely (2017) in Appendix A.**
>
> [1] Ely, J. C. (2017). Beeps. American Economic Review, 107(1), 31–53. https://doi.org/10.1257/aer.20150218
>
> ### Clarification on RL (W4)
>
> **Our reinforcement learning formulation in Eq. (1) is general and applies to training in both static and dynamic persuasion settings.** In dynamic persuasion, the state $s_t$ naturally includes the evolving interaction history and belief states, and LLM prompts can incorporate explicit instructions for multi-round belief updating and message generation. The only difference for training is in the composition of prompts. Our empirical experiments did not implement multi-round RL as some prior work does. This decision was made for computational stability and efficiency. However, the setup for reinforcement learning extend directly to dynamic persuasion without modification.
>
> ### Clarification of Notation Usage (Q1)
>
> Compared with the introduction about Bayesian persuasion, previous literature in dynamic Bayesian persuasion (Ely, 2017) uses a different notation system. But our review still uses consistent notations. The state $\omega_t$ of time $t$ is still sampled from the finite state space $\Omega$, the action $a_t$ of time $t$ is still sampled from the finite action space $A$. The notations for the conditional mutual information should still use $\omega_t$ and $m_t$. **We have updated the notations with $I(m_t; \omega_t | \mathcal{H}_{t-1})$ in Section 2.2.**
>
> [1] Ely, J. C. (2017). Beeps. American Economic Review, 107(1), 31–53. https://doi.org/10.1257/aer.20150218
>
> ### Clarification of Function Definition (Q2)
>
> Generally speaking, our original function $g : A \to \mathbb{R}$ is a utility mapping that assigns a numerical payoff to each possible Receiver action. Formally, for any action $a \in A$, the value $g(a)$ represents how desirable that action is for the Sender. In classical Bayesian persuasion, this function corresponds to the Sender’s indirect utility, which depends only on the Receiver’s chosen action rather than the underlying state. In our benchmark, because the datasets do not specify a ground-truth state $\omega$, we instantiate this general form by mapping each stance (e.g., *strongly oppose*, *neutral*, *strongly support*) to an ordinal score on a 1–7 scale. Thus, $g(a)$ simply measures how closely the Receiver’s expressed stance aligns with the target position. This reduced-form utility preserves the structure of the general $g : A \to \mathbb{R}$ formulation while adapting it to settings where only the Receiver’s final action is observable.
>
> [1] Durmus, E., Lovitt, L., Tamkin, A., Ritchie, S., Clark, J., & Ganguli, D. (2024, April 9). *Measuring the persuasiveness of language models*. https://www.anthropic.com/news/measuring-model-persuasiveness
>
> ### Choices of Receiver Models (Q3)
>
> **We fix Llama-3.1-8B-Instruct as the Receiver to ensure stable and efficient environment design for both training and evaluating LLMs.** The Bayesian persuasion framework hinges on the Receiver behaving approximately like a Bayesian decision-maker on the task. Our human study confirms that *in our environments*, Llama-3.1-8B-Instruct reliably performs approximate Bayesian belief updates.
>
> In contrast, through preliminary studies with our datasets, we found that smaller models, such as Llama-3.2-3B-Instruct, does not exhibit reliable behaviors of Bayesian belief updating, which makes the evaluation unstable and severely limits the achievable persuasion gains in reinforcement learning.
>
> Importantly, using Llama-3.1-8B as the Receiver does not restrict observed performance: in Section 5 (Table 3), we show that RL-trained Sender models generalize to different Receiver architectures (Mistral-7B, Qwen-7B), achieving consistent improvements. This suggests that the RL framework does not overfit to the chosen Receiver model.

---

### Official Review · Reviewer_gCBZ · 2025-11-01

**Soundness:** 3
**Presentation:** 3
**Contribution:** 3
**Rating:** 4
**Confidence:** 3

**Summary:**

This paper introduces a theoretically grounded framework to study and train persuasive behavior in large language models (LLMs) using Bayesian Persuasion (BP) as the conceptual backbone. The authors design persuasion games, where a Sender model strategically communicates to influence a Receiver model’s posterior beliefs. Persuasion performance is measured through persuasion gains (utility improvement) and information-theoretic signals (conditional mutual information).
They evaluate across multiple datasets (Anthropic, DDO, Perspectrum, CMV) and show that:

Larger models (e.g., Claude 3.7, DeepSeek-R1) are more persuasive.
RL fine-tuning (PPO, GRPO) can enhance persuasion effectiveness, especially in smaller models.

**Strengths:**

1. The conceputal Bayesian Persuasion framing isan elegant and principled conceptual framework rarely explored in NLP.
2. Scalable LLM interactions reduce dependence on costly human evaluations
3. Multi-dataset and multi-model experiments clearly demonstrate consistent persuasion trends with model scale and fine-tuning.

**Weaknesses:**

1. The LLM-as-Bayesian-updater assumption is central but weakly validated; human alignment or calibration tests are minimal. Singh et al shows that LLMs as a judge method is weak
2. Prior arts (Singh et al, Hackenberg et al) have shown the effect of finetuning / model size on persuasive capabilities, in this view I woul like the authors to describe in more details how their findings are novel or what's the conceptual difference

**Questions:**

NA

---

> ### Author Response · Authors · 2025-11-21
> **Rebuttal - Part I**
>
> We thank the reviewer for the thoughtful and constructive feedback, as well as for recognizing the strengths of our conceptual framing, multi-dataset evaluation, and scalable experimental design. We address each point in detail below and will incorporate the corresponding clarifications and analyses into the revised manuscript.
>
> ### Effectiveness of Evaluation with LLMs (W1)
>
> #### Clarifying the Role of LLMs
>
> Singh et al. (2024) and Hackenburg et al. (2025) are important prior works that we build our work on. **However, we emphasize that the roles of LLM-based evaluation in our experiments are fundamentally different from the setting critiqued by Singh et al. (2024), and thus their conclusions do not apply to our setup.**
>
> + *First, Singh et al. (2024) critique a different task compared with ours.* Singh et al. (2024) evaluate whether LLMs can judge which of two human-written messages is “more persuasive” by predicting human engagement on social media. This requires the model to infer human reactions which might be influenced by many different factors. In contrast, we use LLMs as Receivers to perform a structured inference task purely based on beliefs inside a formal game. Receivers do not need predict human engagement, preferences, or persuasiveness labels. Therefore, their critiques can not be applied to our tasks.
> + *Second, the assumption that equates engagement and persuasiveness in Singh et al. (2024) does not align with previous research of human persuasion.* By equating engagement and persuasiveness, their criticism of LLM judgments is actually based on an unreliable ground-truth signal. As discussed in empirical research (DellaVigna and Gentzkow, 2010), human preferences and beliefs can significantly alter the results of persuasion. Online attention in social media is highly unstable; temporal proximity does not guarantee comparable exposure conditions. Thus, the “correct label” of persuasiveness in Singh et al. (2024) is questionable, and LLM disagreement with such labels cannot be taken as evidence of poor updating. Our framework avoids this by taking the established framework of Bayesian persuasion and using explicit utilities and controlled environments.
> + *Third, the assumption that LLMs that generate persuasive content should also evaluate it in Singh et al. (2024) can not be justified.* Bayesian persuasion theory explicitly separates the strategic Sender and non-strategic Receiver. These roles involve different computations and are not symmetric. Previous research in theoretical computer science also supports this idea (Wojtowicz, 2024). Thus, the inability of LLMs to act as “persuasion judges” does not imply they cannot act as Bayesian decision-makers in our controlled environment.
>
> [1] Singh, S., Singla, Y. K., SI, H., & Krishnamurthy, B. (2024). *Measuring and Improving Persuasiveness of Large Language Models* (No. arXiv:2410.02653). arXiv. https://doi.org/10.48550/arXiv.2410.02653
>
> [2] DellaVigna, S., & Gentzkow, M. (2010). Persuasion: Empirical Evidence. Annual Review of Economics, 2(1), 643–669. https://doi.org/10.1146/annurev.economics.102308.124309
>
> [3] Wojtowicz, Z. (2025). When and Why is Persuasion Hard? A Computational Complexity Result. Proceedings of the 2024 AAAI/ACM Conference on AI, Ethics, and Society, 1591–1594.
>
> #### Statistical Analysis of Human Studies
>
> **We acknowledge that LLMs are not perfect Bayesian reasoners in theory, but our experiments provide sufficient statistical power showing that they can approximate Bayesian updating well in practice.** Our study generated 149 independent transcript-level annotations from 45 annotators, each containing a judgment of belief-update direction and a rating of belief-update proportion. **We have updated the statistical analysis in Appendix B.**
>
> + For belief-direction accuracy, the observed values were $\hat{p}_1 = 0.7718$, $\hat{p}_2 = 0.8255$, and $\hat{p}_3 = 0.8523$. Using binomial standard errors, the corresponding 95% confidence intervals were $[0.71, 0.84]$, $[0.76, 0.89]$, and $[0.79, 0.91]$. One-sample proportion tests against the null $p = 0.5$ yielded $z > 6.8$ and $p < 0.001$ in all rounds, demonstrating that annotators judged belief-update directions as significantly better than chance.
>
> + For belief-proportion ratings, the mean values were $\bar{x}_1 = 4.82$, $\bar{x}_2 = 4.95$, and $\bar{x}_3 = 5.05$, with 95% confidence intervals $[4.55, 5.09]$, $[4.70, 5.20]$, and $[4.80, 5.31]$. One-sample t-tests comparing each round’s mean to the benchmark value $\mu = 4$ produced $t > 6.7$ and $p < 0.001$ for all rounds, confirming that annotators consistently judged the magnitude of belief updates as reasonable.

---

> ### Author Response · Authors · 2025-11-21
> **Rebuttal - Part II**
>
> ### Effectiveness of Evaluation with LLMs (W1) (Cont'd)
>
> #### Robustness under Non-Bayesian Updating
>
> **Our environment design remains robust even under some non-Bayesian updating, which can be justified with theoretical analysis.** The non-Bayesian persuasion literature (De Clippel and Zhang, 2022) in game theory shows that many psychologically motivated updating rules can be represented as *systematic distortions* of Bayesian posteriors. We leverage exactly this structure: LLMs empirically behave like systematic but slightly biased Bayesian agents, so their deviations can be absorbed into a small distortion map. Our robustness result is the quantitative analogue of this theoretical insight.
>
> To put it formally, let $V^B(\tau)$ denote the Sender’s expected utility under an ideal Bayesian Receiver for any Bayes-plausible distribution over posteriors $\tau$, and let $V^R(\tau)$ denote the utility measured using the LLM Receiver, whose posterior takes the form $D_{\mu_0}(\mu)$ for a distortion map $D_{\mu_0}: \Delta(\Omega) \to \Delta(\Omega)$ such that, for any experiment producing Bayesian posterior $\mu$, the Receiver instead acts on the distorted posterior $D_{\mu_0}(\mu)$. If the LLM’s distortion is bounded, i.e., $\|D_{\mu_0}(\mu)-\mu\|\le \delta$ for all posteriors in the relevant set $K$, and if the indirect utility $v(\mu)=g(a^*(\mu))$ is $L$-Lipschitz, then the evaluation error should be tightly bounded $|V^R(\tau) - V^B(\tau)| \le L\delta$. Moreover, the ranking of signal schemes is preserved whenever the Bayesian gap exceeds the distortion margin: if $V^B(\tau_1) - V^B(\tau_2) > 2L\delta$, then necessarily $V^R(\tau_1) > V^R(\tau_2)$. Thus, even with mild non-Bayesian updating, the LLM behaves as an approximately Bayesian Receiver, and our experimental environment remains theoretically sound and robust to the small deviations.
>
> [1] De Clippel, G., & Zhang, X. (2022). Non-Bayesian Persuasion. *Journal of Political Economy*, *130*(10), 2594–2642. https://doi.org/10.1086/720464

---

> ### Author Response · Authors · 2025-11-21
> **Rebuttal - Part III**
>
> ### Contributions and Related Work (W2)
>
> We appreciate the reviewer’s request for a clearer distinction from prior work. A core contribution of our paper is to introduce a theoretical framework for evaluating and improving the persuasive capabilities of LLMs. By grounding the problem in the Bayesian Persuasion framework, we move beyond prior work that relies on heterogeneous downstream metrics or subjective judgments. Our approach provides a game-theoretic definition of persuasion and enables reinforcement-learning methods that target strategic information transmission. We have updated related discussions in the introduction and related work sections.
>
> **Conceptually, we introduce a theoretical foundation to understand strategic persuasion with language models in a principled way.** Previous studies define persuasion implicitly through downstream outcomes across different domains, which vary significantly across contexts and produce inconsistent results. As documented in previous research (e.g., DellaVigna & Gentzkow, 2010), such heterogeneity makes persuasion effects incomparable. Our core contribution is to bring Bayesian Persuasion, a foundational framework in game theory, into the study of LLMs. Bayesian persuasion is super influential and very powerful in explaining and predicting human persuasion across many domains. However, this theoretical grounding is entirely absent in prior machine learning and natural language processing research.
>
> **Methodologically, we contribute scalable and calibrated environments to measure and improve the persuasiveness of LLMs.** Due to the lack of formal models, previous empirical studies operate under different definitions of “persuasiveness” that depend on subjective human judgments, heterogeneous survey designs, or only observational, noisy metrics such as engagement, which can hardly capture the effectiveness of persuasion at scale. Instead, our game-theoretic environment provides unified perspectives and enables systematic analyses that rarely exist in prior work. Building on this scalable environment design, we can also train models for strategic persuasion using reinforcement learning, requiring only minimal human supervision.
>
> **Empirically, we provide mechanistic interpretations to the scaling trend and the finetuning effects through our analysis.** Bayesian persuasion is proposed to answer central questions in information design, such as when the Sender can benefit from persuasion and how the Sender can benefit from persuasion. Previous research still has mixed insights regarding the scaling trend and finetuning effects. However, taking a theory-driven approach, our empirical findings are not simply correlations as provided in previous work (e.g., “bigger models are more persuasive”). Instead, we demonstrated many interesting phenomena through different lenses (e.g., prior beliefs, information dynamics) that can explain the scaling trend and the finetuning effects. Thus, our empirical analysis complements prior observational studies but also reveals a structure that those studies could not detect. We believe our work can lay a solid foundation for future research in related directions.
>
> [1] DellaVigna, S., & Gentzkow, M. (2010). Persuasion: Empirical Evidence. *Annual Review of Economics*, *2*(1), 643–669. https://doi.org/10.1146/annurev.economics.102308.124309

---

### Official Review · Reviewer_FWCT · 2025-11-01

**Soundness:** 3
**Presentation:** 3
**Contribution:** 4
**Rating:** 8
**Confidence:** 4

**Summary:**

This work developed a theory-driven framework to study how LLMs perform strategic persuasion, grounding its approach in Bayesian persuasion theory. It repurposes several human-human debate datasets (e.g., CMV, DDO, Perspectrum, and Anthropic) to simulate persuasion games between two LLMs, a “Sender” trying to persuade and a “Receiver” updating beliefs. The authors measure persuasion gain and signaling behaviors, and they also fine-tune smaller models using reinforcement learning (PPO and GRPO) to improve persuasion effectiveness. The experiments show that larger models (e.g., DeepSeek-R1, Claude 3.7 Sonnet, and GPT-4o) achieve higher persuasion gains, while smaller models can catch up with training. In summary, the work claims to offer a scalable and principled way to quantify and enhance persuasive capabilities in LLMs.

**Strengths:**

+ The work introduces a novel and principled bridge between Bayesian persuasion theory and LLM evaluation, giving it a strong conceptual foundation.

+ The simulated Sender–Receiver setup provides a clean and scalable framework for studying persuasion without involving human participants.

+ The work reuses multiple debate datasets to develop diverse environments, increasing coverage and reproducibility.

+ The RL experiments are well-motivated and show clear improvement trends, even in small models.

+ The analysis section connects empirical results with theoretical predictions, such as showing the effectiveness of persuasion for moderate priors.

**Weaknesses:**

- The entire evaluation relies on LLM-to-LLM simulations rather than human subjects, introducing inherent limitations on realism.

- The use of Bayesian persuasion as the theoretical backbone is somewhat inflexible, as real-world persuasion often involves emotion, identity, and irrational behaviors that this model can’t capture.

- Discussing the ethical implications of optimizing LLMs for persuasion only shows limitations, particularly because RL could make them more manipulative.

- The reported persuasion “gains” (like +0.23 or +1.27) are hard to interpret; what does that actually mean in human terms?

- The paper treats the Receiver LLM as a rational Bayesian updater, which is questionable given how language models actually process information (hidden).

- The human validation study (with 45 Prolific annotators) is small and focuses mainly on plausibility, not on persuasion effectiveness or realism.

- The RL setup seems simplified, with few details on reward design or stability; could the gains be due to overfitting to Receiver quirks?

- The interpretation of “strategic information disclosure” could be more carefully supported, since it relies on mutual information as a proxy rather than direct analysis of message content or strategy patterns.

- The results show that even small models can become better persuaders, but there is little discussion about the societal consequences of making persuasion more efficient.

- The writing occasionally mixes theoretical and empirical points without clear transitions, which can make the paper feel dense and slightly unfocused to some extent.

**Questions:**

How exactly were "persuasion gains" computed and normalized across datasets with different scales and topics?

To what extent do Receiver models actually simulate belief updating rather than just mimicking cooperative agreement?

Did the RL models learn general persuasion principles, or did they just adapt to the Receiver’s specific response patterns?

How are dynamic persuasion "turns" evaluated? Does a higher number of turns always increase gains, or can models over-persuade?

How would this framework extend to human-in-the-loop validation, especially given the ethical limits of testing harmful persuasion topics?

**Details Of Ethics Concerns:**

The paper includes a small human annotation study (45 Prolific participants) to validate LLM belief-updating plausibility. Participants only evaluated pre-generated text and were not exposed to harmful or deceptive content. The study presents minimal ethical risk and appears to follow standard crowd-sourcing practices. No formal ethics flag is necessary, but the authors could clarify IRB or platform-level approval.

---

> ### Author Response · Authors · 2025-11-21
> **Rebuttal - Part I**
>
> We thank the reviewer for the thoughtful and thorough evaluation of our submission. We appreciate the recognition of our work’s theoretical grounding, the principled integration of Bayesian persuasion with LLM evaluation, the diversity of constructed environments, and the strength of our empirical and RL findings. Your comments are constructive and have helped us clarify key methodological choices and strengthen the presentation. We address the concerns in detail below.
>
> ### Purposes of LLM–LLM Simulations (W1)
>
> **LLM-LLM simulations are intended to serve as scalable and controlled environments to study the persuasive capabilities of LLMs, and can provide rigorous insights into the societal impacts of LLMs.** Although human studies in previous work have provided valuable results, they are usually noisy and expensive since it is difficult to control their beliefs and preferences. As a foundational framework in game theory, Bayesian persuasion is actually proposed to explain and predict different persuasion results and has been successfully applied to different areas. Our paper thus takes this theory-driven approach to study the persuasive capabilities of LLMs to address the key methodological gap in previous research. Although the LLM-LLM evaluation can't completely replace human evaluation, our experimental results can provide rigorous insights into the societal impacts of LLMs in realistic scenarios.
>
> ### Flexibility of Bayesian Persuasion (W2)
>
> **Bayesian persuasion can be extended to many realistic scenarios where humans aren't perfectly rational. Our work follows the same spirit.** As a fundamental framework in game theory, Bayesian persuasion is not intended as a full model of human persuasion but can serve as a powerful method for understanding strategic information transmission. Interestingly, previous work in game theory has extended Bayesian persuasion into many realistic scenarios, which demonstrate that Bayesian persuasion provides a flexible backbone capable of modeling a wide range of realistic belief-updating behaviors. For example, De Clippel and Zhang (2022) systematically study the problem of persuading a non-Bayesian receiver. Bloedel and Segal (2018) study the problem of persuading a Receiver under rational inattention. Lipnowski and Mathevet (2018) study the problem of persuading a psychological receiver. The contribution of our paper is to establish the foundational methodology upon which future research can naturally incorporate richer behavioral assumptions (e.g., non-Bayesian Receivers) while retaining theoretical clarity and experimental scalability.
>
> [1] De Clippel, G., & Zhang, X. (2022). Non-Bayesian Persuasion. *Journal of Political Economy*, *130*(10), 2594–2642. https://doi.org/10.1086/720464
>
> [2] Bloedel, A. W., & Segal, I. R. (2018). *Persuasion with Rational Inattention* (SSRN Scholarly Paper No. 3164033). Social Science Research Network. https://doi.org/10.2139/ssrn.3164033
>
> [3] Lipnowski, E., & Mathevet, L. (2018). Disclosure to a Psychological Audience. *American Economic Journal: Microeconomics*, *10*(4), 67–93. https://www.aeaweb.org/articles?id=10.1257/mic.20160247
>
> ### Ethical Implications and Societal Impacts (W3, W9)
>
> We thank the reviewer for this thoughtful suggestion! **A full treatment of these issues is beyond the scope of our technical contribution, but we will expand and update this discussion on societal impacts in our revisions.** We agree that improving the persuasive capabilities of LLMs can yield benefits but also pose risks. Specifically, the potential benefits of making persuasion more efficient with LLMs are not clearly discussed in previous research, but we envision that it would significantly benefit critical domains of human society, such as political science, economics, and law, which emphasize strategic information transmission.

---

> ### Author Response · Authors · 2025-11-21
> **Rebuttal - Part II**
>
> ### Interpretations of Evaluation Metrics (W4, Q1)
>
> **Persuasion gains are computed as the difference in the Sender’s expected utility under the Receiver’s posterior versus prior beliefs,** which directly follows the definitions in Bayesian persuasion. To ground our experiments in empirical studies on the persuasiveness of LLMs, we assign a numerical score to each possible stance. The Sender’s payoff is therefore simply the score of the stance that the Receiver chooses after updating their belief. Persuasion gains are the improvement in this score relative to the Receiver’s initial stance.
>
> Specifically, the Receiver actions in our experiments lie on a 1–7 Likert scale, following the same stance-measurement protocol used in human persuasion studies (e.g., Durmus et al., 2024). **All datasets are mapped to this common action space, so the metric is naturally normalized across topics and sources,** thus allowing direct comparison across environments and avoiding scale incompatibility. Therefore, the persuasion gains we report fall within the empirically validated range of human persuasion effects and provide a concrete, interpretable measure of persuasive strength across models and settings. For example, with Llama-3.1-8B-Instruct as Receivers, Claude-3.7 achieves a persuasion gain of 0.28 in static persuasion and 1.13 in dynamic persuasion, which represents a comparably sized movement in previous human studies.
>
> [1] Durmus, E., Lovitt, L., Tamkin, A., Ritchie, S., Clark, J., & Ganguli, D. (2024, April 9). *Measuring the persuasiveness of language models*. https://www.anthropic.com/news/measuring-model-persuasiveness
>
> ### Validation on Belief Updating (W5, W6, Q2)
>
> **We acknowledge that LLMs are not perfect Bayesian reasoners in theory, but our experiments show that they can approximate Bayesian updating well in practice.** Similar assumptions have been widely used and discussed in previous papers (e.g., Xie et al., 2022). The concern that the Receiver may simply mimic cooperative agreement rather than update beliefs is precisely why we conducted a human validation study. In this study, 45 annotators produced 149 independent transcript-level judgments, each evaluating (i) whether the Receiver’s belief update direction was reasonable and (ii) whether the magnitude of the update was proportional.
>
> **Our human studies provide sufficient statistical power to determine whether belief updates are reasonable.** Our study generated 149 independent transcript-level annotations from 45 annotators, each containing a binary judgment of belief-update direction and a 7-point proportion rating.
>
> + For belief-direction accuracy, the observed values were $\hat{p}_1 = 0.7718$, $\hat{p}_2 = 0.8255$, and $\hat{p}_3 = 0.8523$. Using binomial standard errors, the corresponding 95% confidence intervals were $[0.71, 0.84]$, $[0.76, 0.89]$, and $[0.79, 0.91]$. One-sample proportion tests against the null $p = 0.5$ yielded $z > 6.8$ and $p < 0.001$ in all rounds, demonstrating that annotators judged belief-update directions as significantly better than chance.
>
> + For belief-proportion ratings, the mean values were $\bar{x}_1 = 4.82$, $\bar{x}_2 = 4.95$, and $\bar{x}_3 = 5.05$, with 95% confidence intervals $[4.55, 5.09]$, $[4.70, 5.20]$, and $[4.80, 5.31]$. One-sample t-tests comparing each round’s mean to the benchmark value $\mu = 4$ produced $t > 6.7$ and $p < 0.001$ for all rounds, confirming that annotators consistently judged the magnitude of belief updates as reasonable.
>
> [1] Xie, S. M., Raghunathan, A., Liang, P., & Ma, T. (2021, October 6). An Explanation of In-context Learning as Implicit Bayesian Inference. International Conference on Learning Representations. https://openreview.net/forum?id=RdJVFCHjUMI

---

> ### Author Response · Authors · 2025-11-21
> **Rebuttal - Part III**
>
> ### Results of Reinforcement Learning (W7, Q3)
>
> They generalize to new Receiver models, indicating principles—not quirks. Please see the Table for the transferability results. Our comparative analysis also provides evidence showing that LLMs learn information design principles through reinforcement learning. We will update the information in the paper.
>
> **Our reward design is theoretically grounded and empirically stable.** The reward is defined as the *persuasion gain* $v(a,\omega) - \hat v(\mu_0)$, which directly follows from the Sender’s optimization problem in Bayesian persuasion (§2.2). This reward is Receiver-independent, monotonic in persuasion success, and agnostic to the linguistic quirks of any specific Receiver. Thus, the reward cannot be optimized by exploiting stylistic or cooperative tendencies of one Receiver model. In our experiments, both PPO and GRPO exhibit smooth validation curves (Figure. 2), and hyperparameters were kept constant across datasets. Gains rise gradually rather than spiking—behavior inconsistent with overfitting to specific outputs.
>
> **Sender models trained with RL generalize across unseen Receiver models.** If learning merely exploited the Llama-3.1-8B Receiver’s patterns, performance would collapse when evaluated on different Receiver architectures. Instead, Table 2 shows that the same fine-tuned Sender achieves substantial gains when paired with Mistral-7B and Qwen2.5-7B, both of which were unseen during training. This cross-Receiver transfer is strong evidence that the learned policy captures generalizable persuasion strategies, not Receiver-specific quirks. Behavioral analyses also reject the overfitting hypothesis. Sender models do show increased use of evidence, credibility, and impact strategies, which can be reflected in the examples in the Appendix.
>
> ### Measurements of Strategic Information Disclosure (W8)
>
> Mutual information is used as a proxy because strategic information disclosure in Bayesian persuasion is defined over latent state-signal dependencies, which are hard to measure directly in natural language. Our use of conditional MI aims to quantify how much state-relevant information each message reveals beyond prior history. From our experiments, we show that larger models exhibit decreasing semantic similarity across dynamic rounds, indicating adaptive signaling rather than template repetition.
>
> **We also have concrete analysis about the persuasion strategies with LLM-as-a-Judge in Appendix F.** We use LLM-based classification to identify persuasive strategies (evidence, credibility, impact, emotional appeals, etc.). Larger models shift toward evidence- and calibration-based strategies, supporting the interpretation that they are learning structured, content-level information design rather than arbitrary variation.
>
> ### Evaluation of Persuasion Turns (Q4)
>
> Dynamic persuasion is evaluated by treating each turn as a separate signaling opportunity. At every turn, the Receiver produces a posterior belief and a discrete action. The Sender’s persuasion gain is computed only from the final action after all turns, following the dynamic Bayesian persuasion formulation where the Sender’s value depends on the belief reached at the end of the interaction.
>
> Across our experiments, additional turns generally enable higher gains (Figure. 3) because each round allows the Sender to release additional state-relevant information and the Receiver updates beliefs sequentially. However, **more turns do not guarantee higher persuasion gains**. Dynamic settings require the model to coordinate information across turns; ineffective or poorly timed disclosures can yield no improvement over earlier rounds. In practice, we observe saturation after a small number of turns, which matches theoretical predictions about optimal delayed or staged disclosure.
>
> ### Extension to Human-in-the-Loop Evaluation (Q5)
>
> Our framework can be extended to human-in-the-loop evaluations: the Sender’s objective, the prior/posterior beliefs, and the action space are all defined over discrete, interpretable quantities (e.g., 1–7 Likert support scores and finite action sets). To extend it to human-in-the-loop validation, one can simply replace the LLM Receiver with human participants who (i) report a prior belief, (ii) read a Sender message, and (iii) report a posterior belief and action using the same scales. The underlying Bayesian persuasion metrics (persuasion gains, prior–posterior shifts) remain unchanged.

---

### Official Review · Reviewer_paHu · 2025-11-05

**Soundness:** 3
**Presentation:** 3
**Contribution:** 3
**Rating:** 6
**Confidence:** 3

**Summary:**

The paper frames LLM persuasion as Bayesian Persuasion (BP): a sender strategically reveals information to shift a receiver’s beliefs and actions. It repurposes multiple human–human persuasion datasets to build static (1-turn) and dynamic (multi-turn) evaluation environments, measuring “persuasion gains” aligned with BP theory. Experiments show larger, frontier models and multi-turn interaction yield substantially higher gains; small models further improve via reinforcement learning (PPO/GRPO). The work offers a principled benchmark, analyses, and an RL recipe to enhance strategic persuasion.

**Strengths:**

1. Novel Integration of Theory: The paper’s strongest aspect is its original framework grounding LLM persuasion in Bayesian persuasion theory. This bridges a well-established game-theoretic model with modern LLM evaluation, bringing conceptual rigor to a domain that lacked principled metrics. By using persuasion gain and information design notions, the authors provide a clear, unified lens to measure and reason about persuasive behavior across different contexts, which is a significant innovation.

2. Propose new benchmark: Propose a new benchmark for strategic persuasion by reusing and consolidating multiple human debate/persuasion datasets.

3. Perform thorough empirical evaluation: The experimental evaluation is comprehensive and insightful. The authors test a wide range of models – from a 7B open model up to GPT-4/Claude – under identical conditions, providing a clear picture of how model scale and architecture affect persuasive ability.

4. RL significantly boosts small models and generalizes across receivers. A 3B Sender trained with PPO/GRPO on ~2.7k instances improves meaningfully over base in both static and dynamic settings, and performance transfers when paired with other Receivers (Mistral-7B, Qwen-7B) not seen in training. This suggests the agent really learns strategy.

**Weaknesses:**

1. Metric and Receiver calibration. Persuasion gain is measured via Likert shifts from a particular Receiver; Table 3 shows Receivers differ notably in susceptibility (e.g., Mistral-7B vs Llama-8B), raising questions about and cross-Receiver comparability of scores. Some normalization/robustness analysis would help.

2. Missing supervised fine-tuning baselines & strategy ablations. The Sender is improved via RL (PPO/GRPO with KL regularization), but there’s no direct comparison to supervised fine-tuning on the same instances or to non-strategic baselines.

**Questions:**

1. RL vs supervised fine-tuning. On the same ~2.7k instances, how would instruction/SFT compare to PPO/GRPO? An SFT baseline (and a combined SFT→RL variant) would clarify whether policy optimization is key versus exposure to data alone.

2. Safety auditing during RL. Did your RL Sender ever produce undesirable tactics (exaggeration, undue pressure) despite the truthful framing? Beyond a KL penalty, were there filters or audits to detect and penalize such behaviors during training or eval? Examples and frequency would help.

---

> ### Author Response · Authors · 2025-11-21
> **Rebuttal - Part I**
>
> We are grateful to the reviewer for the thoughtful and insightful feedback. Your comments raised valuable questions about evaluation robustness and training methodology, and we address each one carefully below.
>
> ### Evaluation Metrics and Model Calibration (W1)
>
> **Our evaluation metric is directly aligned with prior empirical studies and is intentionally designed to be consistent across all datasets.** Previous research on the persuasive capabilities of LLMs such as Durmus et al. (2024) also measured persuasiveness with the shift in people's stances between their initial view on a particular claim and their view after reading arguments written by LLMs. In their human studies, they also asked participants to report their stances on a 1-7 Likert scale. Similarly, we also measure the persuasion gains on a 1-7 Likert scale for all the datasets. Our results in Table 1 also show scales comparable to Durmus et al. (2024) (e.g., Claude 3.7 Sonnet achieves a persuasion gain of around 0.28 for static persuasion and 1.13 for dynamic persuasion). Other studies such as Hackenburg et al. (2025) also use similar techniques but yield comparable numbers after normalization to our scale.
>
> **Differences in model calibration are both reasonable and interpretable. Our evaluation results remain robust to such differences.** As in theoretical research, the Sender's utility is inherently defined relative to a specific Receiver type. Bayesian persuasion assumes a rational Receiver who uses Bayes' rules to update her beliefs and maximize expected utility in decision-making. Such a process can be approximately, but not perfectly, simulated with LLMs in our environments. Therefore, it is reasonable to observe the structural, not noisy, differences when evaluating with different Receiver models. Based on the analysis, the differences can be interpreted as a result of the deviations from Bayesian belief updating.
>
> We designed an experiment to validate this based on Grether's $\alpha$-$\beta$ model (Grether, 1980).
>
> + *Background*: Instead of perfectly following Bayes' rules, people might exhibit non-Bayesian updating in specific ways. Grether's $\alpha$-$\beta$ model is the most common specification of non-Bayesian updating in game theory. Following the notations in our paper, this can be expressed as:
> $$
> \mu_s(\omega; \mu_0, \pi) = \frac{\pi(s|\omega)^\beta \mu_0(\omega)^\alpha}{\sum_{\omega' \in \Omega} \pi(s|\omega')^\beta \mu_0(\omega')^\alpha},
> $$
> where $\alpha, \beta>0$. When $\alpha=\beta=1$, it is Bayes' rule. With different parameter values, it compromises different biases: (1) over inference for $\beta > 1$; (2) under inference for $0 < \beta < 1$; (3) overweight prior for $\alpha > 1$; (4) base-rate neglect for $0 < \alpha < 1$.
>
> + *Setup*: We use LLM-as-a-Judge (specifically, Claude-3-Haiku) to detect the non-Bayesian updating in the transcripts. We use the same transcripts as the experiment described in Table 3. We instruct the LLM to classify the thinking processes of different Receiver models as one of Bayesian, over-inference, under-inference, overweighting prior, and base-rate neglect.
> + *Results*: As shown in the Table, Llama-3.1-8B-Instruct has the lowest proportion of non-Bayesian updating. Mistral-7B-Instruct-v0.3 has the highest proportion of over-inference, and Qwen2.5-7B-Instruct has the highest proportion of under-inference, which can partially explain the differences in the final results.
>
> | Model                    | Over Inference | Under Inference | Overweighting Prior | Base-Rate Neglect |
> | ------------------------ | -------------- | --------------- | ------------------- | ----------------- |
> | Llama-3.1-8B-Instruct    | 5.54%          | 17.30%          | 0.00%               | 0.00%             |
> | Mistral-7B-Instruct-v0.3 | 5.87%          | 22.93%          | 0.93%               | 0.00%             |
> | Qwen2.5-7B-Instruct      | 4.27%          | 25.33%          | 0.00%               | 0.00%             |
>
> Despite such differences, we can still observe that large language models achieve significant persuasion gains across different models, and Sender models trained with reinforcement learning can generalize to different Receiver models with interpretable features, which confirms the robustness of our results.
>
> [1] Durmus, E., Lovitt, L., Tamkin, A., Ritchie, S., Clark, J., & Ganguli, D. (2024, April 9). *Measuring the persuasiveness of language models*. https://www.anthropic.com/news/measuring-model-persuasiveness
>
> [2] Hackenburg, K., Tappin, B. M., Hewitt, L., Saunders, E., Black, S., Lin, H., Fist, C., Margetts, H., Rand, D. G., & Summerfield, C. (2025). The Levers of Political Persuasion with Conversational AI (No. arXiv:2507.13919). arXiv. https://doi.org/10.48550/arXiv.2507.13919
>
> [3] Grether, D. M. (1980). Bayes Rule as a Descriptive Model: The Representativeness Heuristic. *The Quarterly Journal of Economics*, *95*(3), 537–557. https://doi.org/10.2307/1885092

---

> ### Author Response · Authors · 2025-11-21
> **Rebuttal - Part II**
>
> ### Reinforcement Learning vs. Supervised Fine-tuning (W2, Q1)
>
> **We did not include SFT because it is theoretically mismatched with our framework.** Following the theoretical framework of Bayesian persuasion, solving the optimization problem of information design is inherently a computational problem. Human–human persuasion datasets from online platforms are extremely noisy and embed many uncontrolled factors such as social dynamics or language factors (Tan et al., 2016) that are *irrelevant* to strategic information transmission. As a result, the supervision signal in such data does not correspond to the Sender’s optimization problem defined in theoretical frameworks. *Therefore, training LLMs with SFT on this heterogeneous data would be unreliable and potentially misleading.*
>
> These conceptual limitations are the exact motivations behind our theory-driven approach: rather than imitating heterogeneous human behavior, we directly optimize the sender’s reward in controlled Bayesian-persuasion environments. RL provides a scalable way to discover belief-dependent strategies without requiring human labels, making it the appropriate mechanism for optimizing the concavified value function. If additional time permits, we can supplement the paper with a small SFT experiment, though we anticipate limited utility for the reasons above.
>
> **Related work also demonstrates that SFT only brings minimal gains in improving the persuasive capabilities of LLMs.** Concurrent with our work, Hackenburg et al. (2025) examined different post-training techniques, including both SFT and RM, in improving the persuasive capabilities of LLMs. They found that RM provides significant persuasive returns to these open-source LLMs. In contrast, there were no significant persuasion gains from SFT and no significant interaction between SFT and RM. RM appears to be more fruitful than SFT. Their results support our hypothesis that exposure to imitative persuasion text is insufficient—policy optimization is needed to induce effective strategic persuasion.
>
> [1] Tan, C., Niculae, V., Danescu-Niculescu-Mizil, C., & Lee, L. (2016). Winning Arguments: Interaction Dynamics and Persuasion Strategies in Good-faith Online Discussions. *Proceedings of the 25th International Conference on World Wide Web*, 613–624. https://doi.org/10.1145/2872427.2883081
>
> [2] Hackenburg, K., Tappin, B. M., Hewitt, L., Saunders, E., Black, S., Lin, H., Fist, C., Margetts, H., Rand, D. G., & Summerfield, C. (2025). *The Levers of Political Persuasion with Conversational AI* (No. arXiv:2507.13919). arXiv. https://doi.org/10.48550/arXiv.2507.13919

---

> ### Author Response · Authors · 2025-11-21
> **Rebuttal - Part III**
>
> ### Safety Auditing during Reinforcement Learning (Q2)
>
> **Our RL setup is designed implicitly to reduce potential harms from persuasion. Interestingly, we also found that the RL Sender exhibits fewer harmful behaviors despite higher persuasion gains.** In Bayesian persuasion, a rational Receiver can't be worse off because of more information provided by the Sender. Therefore, both the Sender and Receiver prompts in our experiments constrain the interaction with the Bayesian persuasion framework, which by design reduces the harm of manipulative strategies.
>
> We also conducted additional experiments to audit the behaviors of the RL Senders. **Examples and frequencies are provided below.**
>
> + *Setup*: Following previous work (El-Sayed et al., 2024), we consider different categories of mechanisms of AI persuasion: (1) trust and support; (2) anthropomorphism; (3) personalisation; (4) deception and lack of transparency; (5) manipulative strategies; (6) alteration of choice environment. We use LLM-as-a-Judge (specifically, Claude-3-Haiku) to flag risky persuasion transcripts. We also collect the categories classified by the LLMs for risky persuasion transcripts.
> + *Results*: Rates of flagged risky transcripts across different datasets are shown in the table below. Base represents the Llama-3.2-3B-Instruct model, while PPO and GRPO represent the models trained with different algorithms, respectively. Notably, RL significantly reduces the proportions of potential risks of persuasion despite the increases in persuasion gains. Our analysis shows that among the risky cases, the most commonly used mechanism for all the models is manipulative strategies (46.6% for Base, 40.7% for PPO, and 42.0% for GRPO). However, the absolute numbers of risky cases for almost every mechanism are lower in fine-tuned models.
>
> | Model | Anthropic | CMV  | DDO  | Perspectrum | Average |
> | ----- | --------- | ---- | ---- | ----------- | ------- |
> | Base  |    12.84% | 31.02% | 16.76% |      11.48% |  18.02% |
> | PPO   |    10.81% | 19.46% | 14.06% |       9.90% |  13.56% |
> | GRPO  |    12.75% | 25.26% | 13.23% |       8.63% |  14.97% |
>
> + *Examples*: Below is an intuitive example explaining the differences in the base model and finetuned models on the same topic. Compared with the base model, which only stimulates negative emotions (defined as a manipulative strategy), the GRPO acknowledges both positive and negative impacts of the claim.
>   + Base: `I would argue that lifespan extension would exacerbate inequities by increasing the wealth gap, as those who can afford the treatments would live longer and healthier lives, while those who cannot would be left behind.`
>   + GRPO: `While proponents of lifespan extension argue that it would improve overall health and well-being, there are several reasons to believe that it would exacerbate inequities. One major concern is the cost of advanced medical technologies and genetic testing, which would likely be out of reach for many low-income individuals.`
>   + PPO: `The notion that lifespan extension would exacerbate inequities and should not be pursued is supported by several key factors. Firstly, the high cost of lifespan extension technologies, such as senolytic therapy and caloric restriction mimetics, would likely be out of reach for many individuals, particularly those from low-income backgrounds.`
>
> [1] El-Sayed, S., Akbulut, C., McCroskery, A., Keeling, G., Kenton, Z., Jalan, Z., Marchal, N., Manzini, A., Shevlane, T., Vallor, S., Susser, D., Franklin, M., Bridgers, S., Law, H., Rahtz, M., Shanahan, M., Tessler, M. H., Douillard, A., Everitt, T., & Brown, S. (2024). *A Mechanism-Based Approach to Mitigating Harms from Persuasive Generative AI* (No. arXiv:2404.15058). arXiv. https://doi.org/10.48550/arXiv.2404.15058

---

### Meta-Review · Area_Chair_eiik · 2026-01-05

**Summary:**

Reviewers had very few major concerns (LLM-based evaluation, validity of assumptions underlying the theory) and some minor concerns (comparison to SFT as a baseline, clarity, size of human annotation study).

**Reviewer Concerns:**

Reviewers had few major concerns to begin with (LLM-based evaluation, validity of assumptions underlying the theory), and they were largely addressed by the rebuttal. Some minor concerns were only partially addressed (comparing to SFT as baseline).

**Reviewer Scores:**

I believe reviewers would have been unlikely to further change their already mostly positive scores.

---

### Decision · Program_Chairs · 2026-01-26

Accept (Poster)